# CT Rendering and Radiomic Analysis in Post-Chemotherapy Retroperitoneal Lymph Node Dissection for Testicular Cancer to Anticipate Difficulties for Young Surgeons

**DOI:** 10.3390/jimaging9030071

**Published:** 2023-03-17

**Authors:** Anna Scavuzzo, Pavel Figueroa-Rodriguez, Alessandro Stefano, Nallely Jimenez Guedulain, Sebastian Muruato Araiza, Jose de Jesus Cendejas Gomez, Alejandro Quiroz Compeaán, Dimas O. Victorio Vargas, Miguel A. Jiménez-Ríos

**Affiliations:** 1Instituto Nacional de Cancerologia, Department of Urology, Universidad Autonoma de Mexico-UNAM, Mexico City 14080, Mexico; 2Instituto Nacional de Cancerologia, Department of Biomedical Engineering, Universidad Autonoma de Mexico-UNAM, Mexico City 14080, Mexico; 3Institute of Molecular Bioimaging and Physiology, National Research Council (IBFM-CNR), 90015 Cefalù, Italy

**Keywords:** testicular cancer, radiomics, retroperitoneal surgery

## Abstract

Post-chemotherapy retroperitoneal lymph node dissection (PC-RPLND) in non-seminomatous germ-cell tumor (NSTGCTs) is a complex procedure. We evaluated whether 3D computed tomography (CT) rendering and their radiomic analysis help predict resectability by junior surgeons. The ambispective analysis was performed between 2016–2021. A prospective group (A) of 30 patients undergoing CT was segmented using the 3D Slicer software while a retrospective group (B) of 30 patients was evaluated with conventional CT (without 3D reconstruction). CatFisher’s exact test showed a *p*-value of 0.13 for group A and 1.0 for Group B. The difference between the proportion test showed a *p*-value of 0.009149 (IC 0.1–0.63). The proportion of the correct classification showed a *p*-value of 0.645 (IC 0.55–0.87) for A, and 0.275 (IC 0.11–0.43) for Group B. Furthermore, 13 shape features were extracted: elongation, flatness, volume, sphericity, and surface area, among others. Performing a logistic regression with the entire dataset, *n* = 60, the results were: Accuracy: 0.7 and Precision: 0.65. Using *n* = 30 randomly chosen, the best result obtained was Accuracy: 0.73 and Precision: 0.83, with a *p*-value: 0.025 for Fisher’s exact test. In conclusion, the results showed a significant difference in the prediction of resectability with conventional CT versus 3D reconstruction by junior surgeons versus experienced surgeons. Radiomic features used to elaborate an artificial intelligence model improve the prediction of resectability. The proposed model could be of great support in a university hospital, allowing it to plan the surgery and to anticipate complications.

## 1. Introduction

Germ cell cancer (GCC) represents one of the most common solid neoplasms affecting young adult men aged 18–44 years and its incidence has risen worldwide in the past two decades [1]. During the past 40 years, there has been an improvement in the survival rates and in the curative rate for men diagnosed with testicular cancer, due a multimodal approach to the management of GCC with the integration of surgery, chemotherapy, and radiation. Moreover, its effective management depends on the knowledge of the pattern of the metastatic spread of disease, primarily to the lymph nodes of the retroperitoneum and to the lung and posterior mediastinum. The retroperitoneal lymph nodes are the most frequent site of metastasis in advanced testicular tumors.

The European Association of Urology (EAU) Guidelines on testicular cancer suggest retroperitoneal lymph node dissection (primary) as the primary treatment in (a) high-risk stage IB patients, (b) highly selected non-seminoma patients, (c) patients with a contraindication to adjuvant chemotherapy and are unwilling to accept surveillance, (d) postpubertal teratoma with a somatic malignant component, and (e) metastatic disease after chemotherapy for stage II or III seminomatous or non-seminomatous germ-cell tumors (NSGCT), depending on the tumor size or after lack of response to chemotherapy [2].

The main parts of RPLND nowadays are therefore being carried out in the postchemotherapy situation. About one-third of patients who received chemotherapy for disseminated GCC have residual disease and demand surgery [3]. Despite the advent of effective chemotherapy offering an adjunct to the technically challenging surgery, RPLND remains an essential part of the treatment algorithm for NSGCT. As opposed to primary RPLND, the surgery in the postchemotherapy setting is more cumbersome and requires complementary procedures [4].

The rationale for post-chemotherapy retroperitoneal lymph node dissection (PC-RPLND) is to remove persistent retroperitoneal lymph nodes. Approximately 30–40% of metastatic NSGCTs exhibit residual tumors after first-line chemotherapy that may contain necrosis/fibrosis (40–50%), mature teratomas (20–40%), or viable carcinoma cells (10–20%). Teratomas are resistant to conventional treatments, so a complete surgical resection with a bilateral retroperitoneal lymphadenectomy rather than salvage radiotherapy or chemotherapy is the first choice [3].

Regrettably, modern imaging techniques poorly differentiated residual necrosis/fibrosis, teratoma, or viable cancer after chemotherapy [5] and neither predict whether the residual masses can be successfully resected or not.

Currently, a shift in the imaging field is taking place: with new interest from the qualitative interpretation of medical imaging to an emphasis on the extraction of quantitative information from medical imaging (namely, radiomics). Radiomics refers to the extraction and analysis of large numbers of advanced quantitative imaging features (radiomic features—RF) from medical images using high throughput methods. Radiomics has two main arms based on how imaging information is transformed into mineable data: handcrafted radiomics and deep learning [6]. Radiomics is an attractive research topic in uro-oncology [7]. Volume rendering is a set of computer methods to obtain an image projection; rendered computerized tomography is not sufficient to obtain a precise visual classification. For this reason, this last task must be complemented with manual or semi-automatic segmentation [8]. The subsequent analysis of radiomic features finally aims at supporting clinical decision making and overcomes the limitations of a purely visual image interpretation [9].

The complete resection of residual retroperitoneal masses in GCC is challenging, even for experienced surgeons, due to their deep anatomic location, desmoplastic reaction, dense peritumoral adhesions, and proximity to major blood vessels or organs. An accurate understanding of the anatomy of the retroperitoneum before the surgical approach is essential for ensuring the achievement of the procedure, especially for young surgeons during the learning curve.

We hypothesized that computerized tomography segment rendering with radiomic extraction could identify whether PC-RP residual masses are resectable during the pre-operative stage by young surgeons.

## 2. Material and Methods

### 2.1. Study Design and Clinical Data

The proposed single-institution and ambispective study included patients diagnosed with NSGCT between 1 January 2016 and 31 October 2021, who had residual retroperitoneal masses after chemotherapy and had undergone PC-RPLND by two surgeons (one training surgeon and one senior surgeon). The study complies with the Declaration of Helsinki, and local ethics committee approval (Instituto Nacional de Cancerologia) was obtained (n. 2020/0123).

For analyses, we selected 30 patients from database of 570 patients who underwent PC-RPLND (retrospective group) and 30 new patients (prospective group) with tumor size from 1 to 13 cm. We considered the retrospective group, as a historical cohort, to compare and to explore the difference of the resectability in those cases studied pre-operatively only by 2D conventional computerized tomography (CT) versus prospective group studied by 3D reconstruction.

The decision to perform pcRPLND was individualized and was taken after discussion in the multi-specialty approach. According to our institution policy, PC-RPLND is performed in patients with NSGCT and a post-chemotherapy retroperitoneal nodal mass more than 1 cm with normal tumor markers. It is also indicated for patients with seminoma and retroperitoneal nodal mass bigger than 3 cm that is positive on a positron emission tomography (PET) combined with the computerized tomography (CT).

Patients underwent clinical examination and testing of serum alpha-fetoprotein (AFP), human chorionic gonadotrophin (hCG), and lactate dehydrogenase (LDH) one week before the PC-RPLND. CT of chest, abdomen, and pelvis were performed four weeks prior to the procedure or after four or six weeks after the beginning of the last cycle of chemotherapy. The testicular primary tumor was removed before the chemotherapy and, in only two cases, the orchiectomy was delayed following chemotherapy; in neither case was the orchiectomy in conjunction with the PC-RPLND.

CS I seminoma, with high risk for recurrence, received two adjuvant courses of carboplatin and CS I non-seminoma adjuvant chemotherapy with bleomycin, etoposide, and platinum (BEP)X 1. CS IIA/ IIB NSGCT have been treated with BEP X 3 or X4 according to risk categories.

For this study, the inclusion criteria were (a) residual nodal size > 1 cm, after frontline cisplatin-based chemotherapy, on CT imaging measured through transverse axial dimension for NSGCT; (b) residual nodal size < 1 cm in patients with intermediate or poor prognosis or pure teratoma in primary orchiectomy specimen; and (c) residual nodal size > 3 cm for seminoma.

Exclusion criteria were absence of contrast-enhanced CT imaging data after chemotherapy; insufficient image quality due to motion artifacts, for example; CT performed outside our institution; and images with tumor size > 13 cm, in the retrospective group. Furthermore, we excluded patients without complete clinical data and pre-operative and intraoperative records or underwent primary RPLND.

Clinical data included: age, prognostic group according to International Germ Cell Cancer Collaborative Group (IGCCCG) classification, serum markers at diagnosis, primary histopathology, serum markers before PC-RPLND, type of PC-RPLND (standard, salvage, desperation, and redo-surgery), histopathology of PC-RPLND, evaluation pre-surgery by an expert surgeon, and outcomes of PC-RPLND (unresectable: yes vs. no). All patients were treated with conventional open surgery. We excluded cases of minimally invasive retroperitoneal lymph node dissection. Transabdominal approach via a midline laparotomy incision was chosen. We performed in all patients, in addition to the resection of the residual tumor identified by imaging, a modification of the surgical template with “split and roll technique”. According to the modified template, we removed all ipsilateral lymph nodes between the level of the renal vessels and the bifurcation of the common iliac artery. As well as resection of the retroperitoneal residual tumor, we carried out for left-sided testis tumors the resection of para-aortic lymph nodes, and, for right-sided ones, paracaval and inter-aortocaval lymph nodes. All cases of PC-RPLND were classified as: *PC-RPLND standard* (after first-line chemotherapy and negative serum markers), *PC-RPLND salvage* (after more lines of chemotherapy and negative serum markers), *desperation RPLND* (applies to patients with persistently elevated or increasing serum tumor markers after primary inductive chemotherapy or after salvage chemotherapy) and *Redo PC-RPLND* (in cases with recurrent or persistent disease after surgery).

### 2.2. Data Analysis

We divided patients into a retrospective (*n* = 30) and prospective (*n* = 30) group. Patients in the retrospective group were evaluated before surgery using a conventional CT approach (without 3D reconstruction), while the patients in the prospective group were evaluated and segmented using 3D Slicer software. Specifically, 3D Slicer has been used to extract radiomic variables that can predict tumor resectability.

Briefly, all pre-operative CT imaging was evaluated by one expert surgeon, with more than ten years of experience in retroperitoneal surgery, and by junior surgeon in training. From the imaging study, they assessed whether masses were resectable. Statistical analyses of clinical data were performed with SPSS (version 25). Continuous variables are presented as median and interquartile range (IQR) and compared using a two-sample t-test or Wilcoxon rank-sum test. Categorical variables are presented as frequency and percentages and compared between groups using chi-squared or Fischer’s exact test.

We used the Pyradiomics python package (Version 3.6) integrated in 3D Slicer for radiomic feature extraction, and SciPy and scikit-learn libraries for data analysis. CT imaging of the 60 patients corresponding to lymph nodes (LN) was segmented and radiomic features for each LN were extracted after standardized image processing. After stepwise feature reduction based on reproducibility, variable importance, and correlation analyses, radiomic features were selected.

A Fisher’s exact test was used to determine if there are non-random associations between preoperative evaluation and surgery results in both groups (using python Scipy library). A two-proportion difference test was performed to determine whether the difference between two proportions of correct association was significant.

## 3. Results

Table 1 summarizes the clinical information of the patients included in this study (*n* = 60). The median age of all patients was 25.50 (IQR = 17–56). The median size of the residual tumor was 89 cm^3^. There were no significant differences between the two subgroups. Figure 1, Figure 2, Figure 3 and Figure 4 are examples of prospective cases in which the tumor analyzed pre-operatively with 3D images is totally resected by a young surgeon. No death occurred during the intraoperative and perioperative period in this study.

Fisher’s exact test was used to determine if there were non-random associations between the preoperative evaluation and surgical outcomes in both the prospective group (denoted as Group A) and retrospective group (denoted as Group B). The test showed *p*-values of 0.13 and 1 for Group A and Group B, respectively. The null hypothesis in both groups is not rejected since there was no statistical significance. Group B offers much more evidence against the null hypothesis than Group A. A two-proportion difference test was then performed to determine whether the difference between the two proportions of the correct association was significant. It showed a *p*-value of 0.009149 (IC 0.1–0.63), with the proportion of correct classification having a *p*-value of 0.645 (IC 0.55–0.87) and 0.275 (IC 0.11–0.43) for the prospective and the retrospective group, respectively.

After a stepwise feature that is reduction-based, thirteen shape features were selected: Elongation, Flatness, LeastAxisLength, MajorAxisLength, Maximum 2D-Diameter Column, Maximum 2D-Diameter Row, Maximum 2D-Diameter Slice, Maximum 3D-Diameter, Mesh Volume, Minor Axis Length, Sphericity, Surface Area, and Surface Volume Ratio.

Using the Pyradiomics package, a logistic regression was performed (with the scikit-learn python library) using the entire dataset (*n* = 60). The algorithm identified 29 true negative cases (VPN), 13 true positive cases (PPV), 11 false negative cases, and seven false positives (Figure 5), with Accuracy: 0.7 and Precision: 0.65. Using a random sample of *n* = 30, the best result had an Accuracy of 0.73 and Precision of 0.83, with a *p*-value of 0.025 for the Fisher’s exact test.

## 4. Discussion

The aim of the study is the prediction of tumor resectability by radiomic segmentation. This topic is critical for the surgeon since retroperitoneal surgery is a very complex procedure; therefore, it is desirable to pre-operatively predict any surgical difficulties. For this reason, we considered two groups of patients: the retrospective group in which patients were evaluated using a conventional CT approach (without 3D reconstruction) and the prospective group in which patients were evaluated and segmented using the 3D Slicer software. The 3D Slicer was also used to identify thirteen radiomic features that may predict tumor resectability. At this point, a logistic regression was performed using the whole data set (retrospective and prospective groups together): 29 true negative cases, 13 true positive cases, 11 false negative cases, and seven false positives were identified with an accuracy of 0.7. The accuracy increased to 0.73 using a random sample of 30 cases. Finally, our statistical analyses showed that there were no non-random associations between the preoperative evaluation and surgical outcomes in both the prospective and retrospective studies. However, the retrospective group offered much more evidence against the null hypothesis than the prospective group. The difference between the proportions test showed that the expert surgeon’s prediction was better by looking at the 3D image than conventional tomography.

The retroperitoneum represents the first metastatic site in 75–90% of NSGCTs of the testis. PC-RPLND represents an integral part of the multimodality treatment in patients with advanced testicular germ cell tumors and it is recommended for residual tumors in the retroperitoneum as soon as possible after chemotherapy. A meaningful benefit regarding progression-free survival and cancer-specific survival was achieved with an immediate surgical approach [10].

The recommendation for the resection of residual masses is based on the observation that, in 35–40% of cases, mature teratoma and, in 10–15%, persistent viable cancer can be found in the PC-RPLND specimen [2,3]. A complete resection of all residual masses during PC-RPLND can be therapeutic, especially in the presence of teratoma, teratoma with somatic transformation, or masses resistant to chemotherapy. Patients with teratoma in the PC-RPLND specimen have excellent disease-free survival of 75–80%, while those with viable GCT have a decreased chance of survival. Surgical approaches are available in the context of open and minimally invasive access [2].

PC-RPLND is a highly complex procedure, compared with standard retroperitoneal surgery, and may require adjunctive procedures, because residual masses can involve adjacent visceral or vascular structures.

Notions of the retroperitoneal anatomy, experience with surgical techniques of the vascular and intestinal structures, and knowledge of the natural history of testicular cancer are imperative for a successful surgery [10].

Conventional cross-sectional imaging and magnetic resonance imaging identify the shape and size of the post-chemotherapy residual retroperitoneal masses, the anatomy of major vessels, and the presence of anatomical variations of relevant structures such as accessory renal arteries, retroaortic veins, or variants of the vena cava or duplicated ureters. Evaluating the relation between the retroperitoneal tumor and abdominal organs requires the reconstruction of both solid organs, such as the spleen, kidney, liver, and pancreas, and hollow organs, such as the stomach and bladder.

When the residual masses are large, there can be an expected involvement of the inferior vena cava (IVC) and the abdominal aorta in about 6–10% and 2% of cases, respectively [10,11]. However, these approaches are not able to recognize whether the residual mass is resectable or if it holds viable tumor cells or fibrosis [12].

The 3D Slicer program has been used in the context of retroperitoneal tumors to determine radiomic variables that can predict their histology. Baessler et al. identified five physical characteristics of tomography, which in an initial model predicted malignancy vs. fibrosis or necrosis with a sensitivity greater than 95% in the pre-test phase, which in its prospective application was adjusted to approximately 85% [13]. The presence of fibrosis or a desmoplastic reaction in the post-chemotherapy residual masses could complicate the surgical resection. The reaction induced by chemotherapy in residual masses often results in a more difficult resection, with firm adherence to the great vessels and adjacent organs. During the surgery, careful handling is required to avoid injury of the ureter, bowel, and vessels.

We propose that CT segment rendering with radiomic feature extraction is essential for supporting experienced surgeons and junior doctors in training during the preoperative stage of the PC-RPLND.

Nowadays, medical image analysis, particularly computed tomography and magnetic resonance imaging, has grown exponentially and helps to plan surgical procedures more precisely, leading to less invasive and more informative diagnoses. Although these tools provide high-resolution two-dimensional images, their ability to describe complex three-dimensional structures is limited [14].

Three-dimensional reconstruction methods offer a better understanding of anatomical complexity, allowing rotations and segmentations in the virtual model [15]. This ability has proven useful for the visualization of complex structures such as congenital heart defects and aneurysms. However, the differences between the real anatomical structures and the interpretation of virtual images in three dimensions are still being studied [14,15,16,17].

It is important to draw up adequate imaging before the surgery, and we argue that CT rendering and radiomic features are superior to conventional imaging during the pre-operative work-out. In this study, we suggest that CT and segment rendering help to predict the resectability of the residual mass and help young surgeons to recognize the anatomy of the tumor. This tool is useful for surgeons in training and for low-volume hospitals to optimize surgical and oncological outcomes. Preoperative planning is the most important part of retroperitoneal surgery, especially for young surgeons in training. With this pre-surgical planning method, intraoperative morbidity, renal loss, vascular injury, and a need for aortic or vena caval resection could be reduced.

Until now, it has not been described as a technique to improve the surgical skill in retroperitoneal surgery. The difference between the proportions test allowed us to confirm that there is a significant difference in the prediction made by the expert surgeon when observing the conventional tomography vs observing the 3D image, the latter being a better tool compared to the former. Our findings show that the radiomic algorithm is more accurate and precise in cases where the post-chemotherapy residual masses are not resectable.

To our knowledge, no previous study has evaluated the prediction of the resectability of retroperitoneal residual masses. However, there are data concerning the need for adjunctive procedures in PC-RPLND, such as nephrectomy, vascular resection or reconstruction, inferior vena cava resection or repair, aortic replacement, duodenectomy, ureteral repair, etc. [15]. Johnson and colleagues described that the dominant mass size and degree of circumferential vessel involvement (>135 for the vena cava and >330 for the aorta) predicted resection or reconstruction [18]. Clinical predictors of the need for additional procedures are risk group, tumor size, final retroperitoneal pathology, and elevated markers [16,19].

We have to keep in mind that PC-RPLND remains a challenging operation with a morbidity of 12% to 32, and 0.8% mortality in experienced specialist centres [20,21,22]. The three-dimensional visualization of the anatomical regions that need to be evaluated for a retroperitoneal lymph node dissection allows us to understand and optimize the procedure, better appreciating the anatomy and planning the surgical route to follow, and, so, improve the perioperative outcome and decrease complications.

CT rendering with radiomic extraction could help predict the result of the surgery more objectively and not dictated by the operator’s experience or skills alone. That said, we believe that radiomic algorithms have the potential to be a useful tool for predicting surgical outcomes in retroperitoneal surgery. Our findings highlight that an artificial intelligence (AI) model is required in the pre-operative planning of advanced testicular tumors compared to the traditional pre-planning by conventional imaging. Currently, AI models and machine learning models are gaining popularity in the field of urology [22,23,24]; our results represent the application of an AI model and the utility of handcrafted radiomics in uro-oncology.

There are limitations to our study. The prediction of resectability depended on surgeon experience, although CT rendering with radiomic extraction allowed the safe resection of retroperitoneal tumor. Further, our findings are based on findings from only one institution, and we were not able to externally validate the model. Therefore, this study provides clues but not sufficient evidence to prove that the proposed AI model can help new surgeons predict if a tumor will be resectable or not. A crucial aspect in the case of testicular cancer is the young age of the patients and, consequently, every attempt should be made to provide curative intent in such cases. For this reason, whether an AI model can improve the prediction of tumor resectability needs to be demonstrated unequivocally. We also encourage other research groups to address this issue.

## 5. Conclusions

While computed tomography allows surgeons to have an overall location of the tumor and could help during surgery planning, CT and segment renderings give us a complete view of the proximity of adjacent vessels and organs. The use of 3D reconstruction adds a more sensitive way of predicting resectability than conventional CT images. By using 3D reconstruction, young surgeons could assess a patient’s preoperative condition, make a surgical strategy, and simulate surgical procedures to ensure a more accurate and safe surgical treatment. The inclusion of radiomic features to build an artificial intelligence model may improve the prediction of resectability in post-chemotherapy RPLND. This tool would be of great support in a teaching hospital, allowing surgery to be planned and complications anticipated, and most importantly, avoiding reaching the point of no return.

## Figures and Tables

**Figure 1 jimaging-09-00071-f001:**
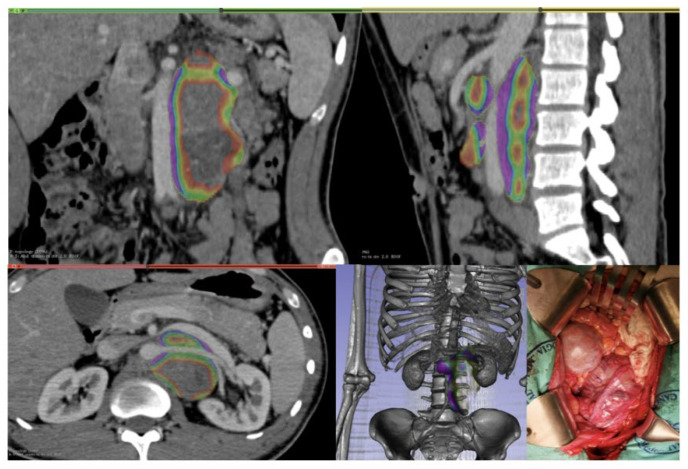
Para-aortic mass, with CT-rendering and 3D reconstruction view precise relationship with aorta and posterior abdominal wall; the mass was completely resected without requirement of resection or reconstruction vascular.

**Figure 2 jimaging-09-00071-f002:**
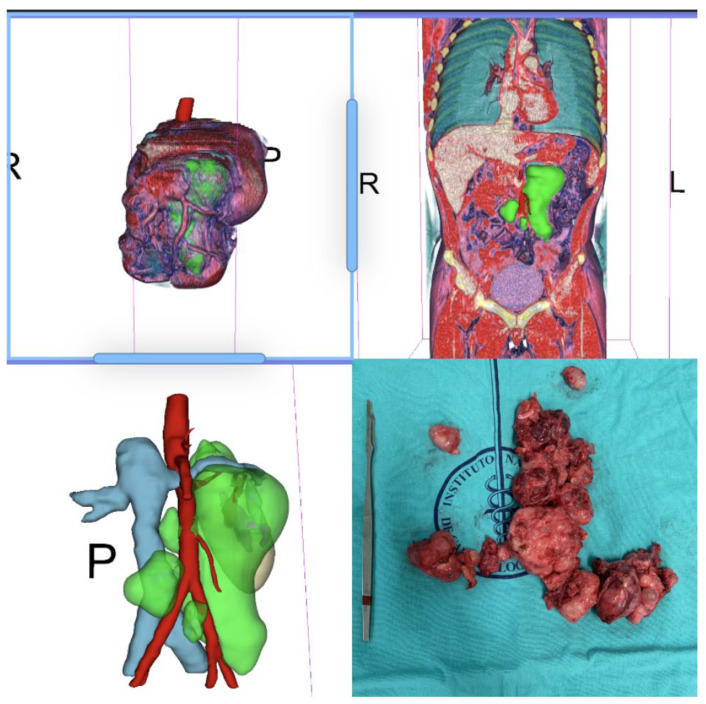
Para and retro-aortic pure post-puberal teratoma, with CT-rendering and 3D reconstruction; the masses were resected without need for adjunctive procedures.

**Figure 3 jimaging-09-00071-f003:**
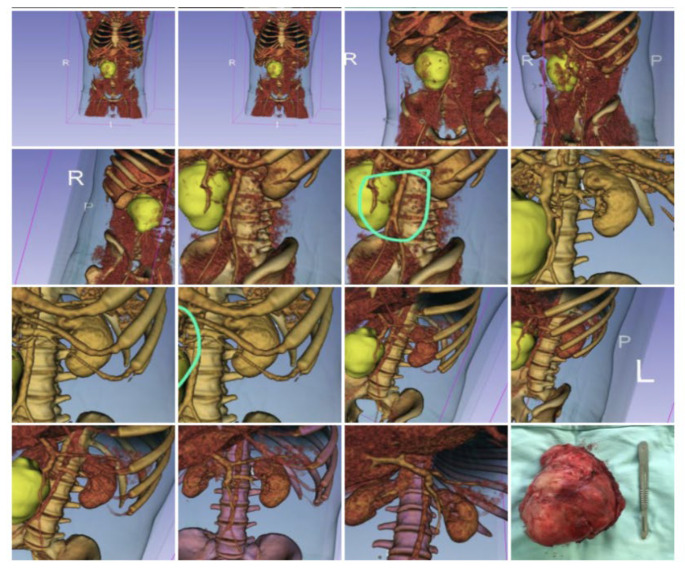
Para and retro-caval Post-chemotherapy Germ Cell Tumor. The mass displaced the cava without infiltration according CT-rendering and 3D reconstruction. It was excised completely without need for vascular surgery. R means right side; L= left side and P = posterior vision; the green circle show the tumor.

**Figure 4 jimaging-09-00071-f004:**
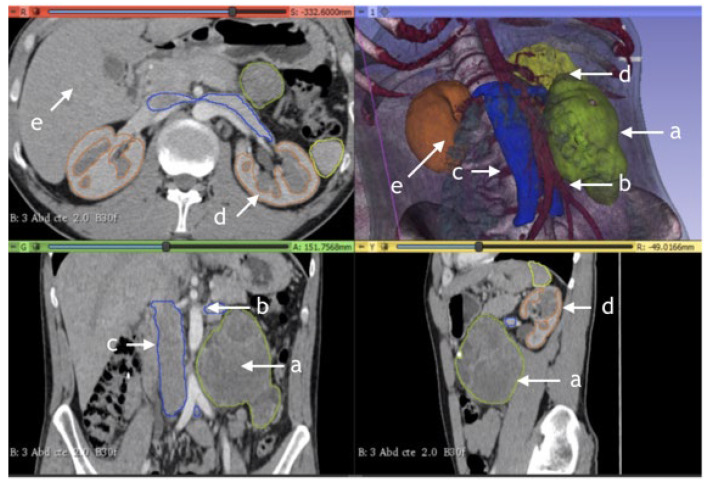
Para-aortic residual mass, post-chemotherapy germ cell tumor. Example of CT-rendered 3D reconstruction: (a) tumor; (b) aorta; (c) v. cava; (d) kidney; and (e) spleen.

**Figure 5 jimaging-09-00071-f005:**
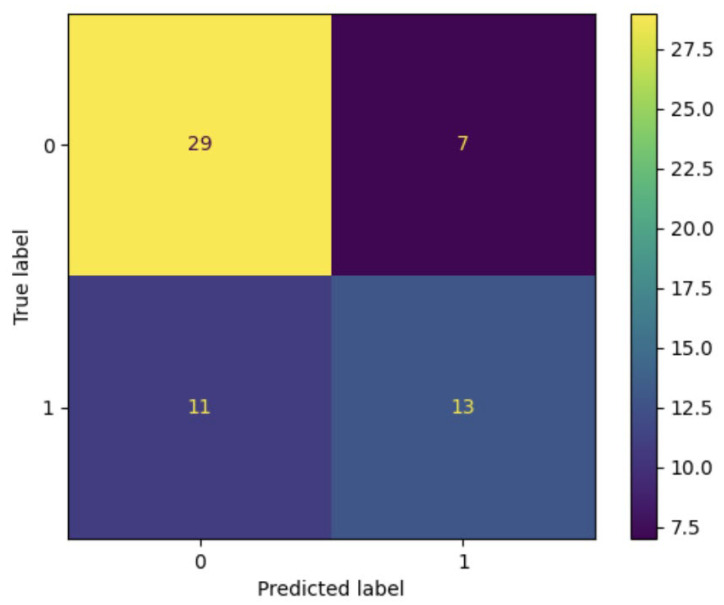
In the whole dataset (*n* = 60), the logistic regression identified 29 true negative cases, 13 true positive cases, 11 false negative cases, and 7 false positives.

**Table 1 jimaging-09-00071-t001:** Clinical information of retrospective and prospective groups.

	Patients *n* = 30 with Conventional CT (Group A, Retrospective)	Patients *n* = 30 with Preoperative Segmentation (Group B, Prospective)	*p*-Value
**Patient age at pcRPLND (years)**	25.21 (17–46)	26.23 (17–56)	0.856
**IGCCCG**			0.321
**Good (*n*)**	8	9	
**Intermediate (*n*)**	14	14	
**Poor (*n*)**	8	7	
**Clinical stage at diagnosis**			0.505
** IA**	1	2	
**IB**	0	1	
**IIA**	3	3	
**IIB**	3	2	
**IIC**	8	3	
**IIIA**	3	2	
**IIIB**	4	9	
**IIIC**	8	8	
**Serum Markers initial**			
AFP	2.712.24	2.165.79	0.635
hCG	13.813.54	3.071.07	0.307
LDH	713.45	766.4	0.024
**Serum Markers postchemotherapy**			
AFP	22.95	21.8	0.505
hCG	0.1	0.2	0.304
LDH	198	178	0.633
**Laterality of primary testicular tumor**			0.302
Right	16	20	
Left	14	10	
**Primary histopathology of testis**			0.065
Seminoma (*n*)	3	3	
Non-seminoma (*n*)	25	23	
Containing teratoma (*n*)	21	16	
Without teratoma (*n*)	9	14	
**Lympho-vascular invasion in the specimen testis**			0.325
Yes	21	18	
No	9	12	
Embryonal > 40% in specimen testis			0.332
Yes	6	7	
No	24	23	
Choriocarcinoma in specimen testis			0.332
Yes	4	6	
No	26	24	
Liver/Bone/Brain			0.332
Metastases at diagnosis	8	7	
**Type of pcRPLND**			0.413
Standard	2	0	
Salvage after chemo	21	19	
Desperation	5	9	
Redo	2	2	
**Pre-surgery pcRPLND evaluation:**			0.39
Non-resectable	7	11	
Resectable	23	19	
**Median Residual Tumor Volume**	8.3 cm^3^	8.5 cm^3^	0.45
	(3 cm^3^–20 cm^3^)	(3 cm^3^–22 cm^3^)	
**pcRPLND unresectable**			0.1195
Yes	10	17	
No	20	13	
**Intraoperative Complications**			0.42
**Vascular injuries **	8	6	
**Organ lesions**	3	1	
**Histopathology pcRPLND**			0.3576
Necrosis-Fibrosis (*n*)	9	11	
Teratoma (*n*)	18	13	
Viable tumor (*n*)	3	6	
Presence of somatic-type malignant transformation	none	none	
**Adjuvant Chemotherapy after pcRPLND**	3	6	0.32

## Data Availability

Not applicable.

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
