# Peer review of "CT Rendering and Radiomic Analysis in Post-Chemotherapy Retroperitoneal Lymph Node Dissection for Testicular Cancer to Anticipate Difficulties for Young Surgeons"

_2313-433X, 2023, doi:10.3390/jimaging9030071_

Round 1
Reviewer 1 Report
Thank you for inviting me to review the article titled “CT rendering and radiomics analysis in post-chemotherapy retroperitoneal lymph node dissection for testicular cancer to anticipate difficulties for young surgeons
The authors evaluate if 3D CT rendering and their radiomics analysis would help predict resectability and identify difference between tumor and fibrosis.
Comments/Criticisms:
Major:
1. I did not understand the point of having a retro and prospective arm in the study.
2. Results and discussions could be organized better. The paper is trying to answer two questions: whether the mass is resectable and the second if the resected mass is actually the tumor or fibrosis. Authors should not assume readers understand that. The sentences “ CatFisher’s exact test was: group A p-value=0.13 and Group B p-value=1.0. The null hypothesis in both groups is not rejected since there was no statistical significance. The retrospective group offers much more evidence against the null hypothesis than the prospective group. The difference between proportions test showed a p-value of 0.009149 (IC 0.1-0.63); Proportion of correct classification, p-value=0.645 (IC 0.55- 0.87) for the prospective group, p-value=0.275 (IC 0.11-0.43) for the retrospective group. “ are not sufficient to explain what it means and authors fail to explain/ explore the results in the discussion.
3. The study might have benefited from actually having more patients for feature identification which most help predict tumor vs. fibrosis/necrosis.
4. While the study mentions how good the AI model can be, it does not mention a single patient from the retrospective arm where surgery was done with a bad outcome using just conventional CT imaging but would have been clearly avoided by the AI model.
Minor:
Introduction :
Word pre is repeated twice. “We hypothesized that CT segments rendering with radiomics extraction could identify whether PC-RP residual masses are resectable during pre-preoperative stage by young surgeons. “
Methods:
Parenthesis needs to be removed after age. “Clinical data included: age, ) prognostic group according International Germ Cell Cancer Collaborative Group (IGCCCG) classification, serum markers at diagnosis, pri- mary histopathology, serum markers before PC-RPLND, type of PC-RPLND (standard, salvage, desperation and redo-surgery), histopathology of PC-RPLND, evaluation pre- surgery by an expert surgeon; outcomes of PC-RPLND (unresectable: yes vs no). “
Figure 4 legend says n =30 , it should be 60.
Author Response
The authors evaluate if 3D CT rendering and their radiomics analysis would help predict resectability and identify difference between tumor and fibrosis.
Comments/Criticisms:
Major:
- I did not understand the point of having a retro and prospective arm in the study.
Author response:
Because only in the retrospective group we value the tumors by with Conventional-CT and we prefer pre-operative studying the patients in prospective arm only by segmentation. the retroperitoneal surgery for metastatic non seminomatous germ cell tumor isn’t frequent; we preferred to carry out a pilot study on a number of patients treated in one year and compared them with a historical group. Not all patients in the retrospective arm showed characteristics similar to the number of patients in the prospective arm, so we decided to reduce the number of observations. In literature the dataset of patients underwent to retroperitoneal surgery in the tertiary hospital included a small number of patients.
- Results and discussions could be organized better. The paper is trying to answer two questions: whether the mass is resectable and the second if the resected mass is actually the tumor or fibrosis. Authors should not assume readers understand that. The sentences “ CatFisher’s exact test was: group A p-value=0.13 and Group B p-value=1.0. The null hypothesis in both groups is not rejected since there was no statistical significance. The retrospective group offers much more evidence against the null hypothesis than the prospective group. The difference between proportions test showed a p-value of 0.009149 (IC 0.1-0.63); Proportion of correct classification, p-value=0.645 (IC 0.55- 0.87) for the prospective group, p-value=0.275 (IC 0.11-0.43) for the retrospective group. “ are not sufficient to explain what it means and authors fail to explain/ explore the results in the discussion.
Author response:
Dear Reviewer, thank you for your suggestion. The paper try to answer only the prediction of resectability of the tumor by radiomic segmentation.
- The study might have benefited from actually having more patients for feature identification which most help predict tumor vs. fibrosis/necrosis.
Author response:
Dear Reviewer, thank you for your suggestion. We have focused this time only on the resectability of the tumour; for the surgeon is important this standpoint; the retroperitoneal surgery is very complex procedure,so it is desirable to pre-operatively predict any surgical difficulties
- While the study mentions how good the AI model can be, it does not mention a single patient from the retrospective arm where surgery was done with a bad outcome using just conventional CT imaging but would have been clearly avoided by the AI model.
Author response: Dear Reviewer, thank you for your suggestion.
Minor:
Introduction :
Word pre is repeated twice. “We hypothesized that CT segments rendering with radiomics extraction could identify whether PC-RP residual masses are resectable during pre-preoperative stage by young surgeons. “
Author response: I corrected the error, thanks
Methods:
Parenthesis needs to be removed after age. “Clinical data included: age, ) prognostic group according International Germ Cell Cancer Collaborative Group (IGCCCG) classification, serum markers at diagnosis, pri- mary histopathology, serum markers before PC-RPLND, type of PC-RPLND (standard, salvage, desperation and redo-surgery), histopathology of PC-RPLND, evaluation pre- surgery by an expert surgeon; outcomes of PC-RPLND (unresectable: yes vs no). “
Author response: Dear Reviewer, thank you for your suggestion.
Figure 4 legend says n =30 , it should be 60. Author response: Ok.I change it, thanks
Reviewer 2 Report
Interesting study all surgeons are aware of the difficulty in resectably. Some short comments:
The authors mention an ambispective study one wonder if the 30 out of 570 pat retrospective was taken from the same years 2016-21, this should be clarified. Had the study ethical approval from the institutional board, this should be mentioned.
The chemotherapy should be mentioned in the text not just in the table.
I believe that presurgical unresectable and the column beneath that surgical unresectable in this it should be say yes first and no under since it easier to understand.
otherwise I think even though it is a negative study it is important to publish.
Author Response
- The authors mention an ambispective study one wonder if the 30 out of 570 pat retrospective was taken from the same years 2016-21, this should be clarified. Had the study ethical approval from the institutional board, this should be mentioned.
Author response: we selected 30 pat from our historical database with the same clinical characteristics of prospective group. The study was approved by the Institutional Review Board (n. 2020/0123).
- The chemotherapy should be mentioned in the text not just in the table.
Author response: Dear Reviewer, thank you for your suggestion. All patients received cisplatin-based chemotherapy ( x3 or x4)
- I believe that presurgical unresectable and the column beneath that surgical unresectable in this it should be say yes first and no under since it easier to understand.
Author response: Dear Reviewer, thank you for your suggestion
Round 2
Reviewer 1 Report
file is attached

Author Response
Reviewer 1
Thank you for inviting me to review the article titled “CT rendering and radiomics analysis in post-chemotherapy retroperitoneal lymph node dissection for testicular cancer to anticipate difficulties for young surgeons.
The authors evaluate if 3D CT rendering and their radiomics analysis would help predict resectability
I commend the authors for taking on the work needed to do the study.
Author response: Firstly, we would like to express our sincere gratitude to the reviewer for your insightful comments. We have revised our manuscript according to your constructive suggestions, and we hope that these revisions can meet your requirements.
Comments:
- The sentences “ CatFisher’s exact test was: group A p-value=0.13 and Group B p-value=1.0. The null hypothesis in both groups is not rejected since there was no statistical significance. The retrospective group offers much more evidence against the null hypothesis than the prospective group. The difference between proportions test showed a p-value of 0.009149 (IC 0.1-0.63); Proportion of correct classification, p-value=0.645 (IC 0.55- 0.87) for the prospective group, p-value=0.275 (IC 0.11-0.43) for the retrospective group. “ . This is not sufficient to explain what it means and authors fail to explain. This is not mentioned in methods either. Who is group A, who is group b?
Author response: Dear Reviewer, we thank you for your suggestion. The whole sentence has been rephrased to make the meaning clearer to the reader. The changes have been underlined in green in the text.
- The first paragraph in the discussion is usually used to summarize the results of the study and open up the discussion about the study results.
Author response: Dear Reviewer, we thank you for your suggestion. We have added a paragraph in the Discussion Section to briefly summarize the results obtained. The changes have been underlined in green in the text.
- While the AI model may seem to help new surgeons to predict if this tumor will be resectable or not, this study has not provided sufficient evidence that this particular model can improve it. An important aspect in the case of testicular cancer one has to remember is that these are young patients and every attempt should be made to provide curative intent in such cases. Thus if an AI model can improve objectivity it has to be proven unequivocally.
Author response: Dear Reviewer, we have added this important consideration in the Discussion Section. Thanks for this valuable comment. The changes have been underlined in green in the text.
Round 3
Reviewer 1 Report
Thank you for improving the explanations and incorporating comments.
No further comments at this time.
Author Response
thanks